# Integrating Social Determinants of Health to Precision Medicine through Digital Transformation: An Exploratory Roadmap

**DOI:** 10.3390/ijerph18095018

**Published:** 2021-05-10

**Authors:** Ik-Whan G. Kwon, Sung-Ho Kim, David Martin

**Affiliations:** 1The Center for Supply Chain Excellence, Saint Louis University, Saint Louis, MO 63108, USA; 2Department of Health Administration, Yonsei University, Seoul 03722, Korea; kimsungho@yonsei.ac.kr; 3The Ancell School of Business, Western Connecticut State University, Danbury, CT 06810, USA; martindg@wcsu.edu

**Keywords:** social determinants of health, precision medicine, AI, machine learning, digital transformation, population health, clinical–community linkages, digital divide, digital literacy

## Abstract

The COVID-19 pandemic has altered healthcare delivery platforms from traditional face-to-face formats to online care through digital tools. The healthcare industry saw a rapid adoption of digital collaborative tools to provide care to patients, regardless of where patients or clinicians were located, while mitigating the risk of exposure to the coronavirus. Information technologies now allow healthcare providers to continue a high level of care for their patients through virtual visits, and to collaborate with other providers in the networks. Population health can be improved by social determinants of health and precision medicine working together. However, these two health-enhancing constructs work independently, resulting in suboptimal health results. This paper argues that artificial intelligence can provide clinical–community linkage that enhances overall population health. An exploratory roadmap is proposed.

## 1. Introduction

The COVID-19 pandemic has altered healthcare delivery platforms from the traditional face-to-face format to online care through digital tools. Unlike the Spanish flu of 1918, which became an international epidemic over the course of a year, COVID-19 spread to every inhabitable continent within weeks, outpacing our health system’s ability to test, track, and contain people with suspected infection [1]. The pandemic also caused a dramatic shift in emergency use, as capacity is limited to manage COVID-19-related patients who used emergency rooms as their first treatment places. As a result, there was a dramatic decline in the use of emergency rooms for injuries not related to COVID-19 episodes [2]. In response to this unprecedented health crisis, the healthcare industry saw the rapid adoption of video conferencing and other digital collaboration tools to provide care to patients, regardless of where patients or clinicians were located, while mitigating the risk of exposure to the coronavirus. Information technologies allow healthcare providers to continue a high level of care for their patients via virtual visits and to collaborate with other providers in the networks. In addition, the era of value-based healthcare, digital innovation, big data, and clinical decision support systems (CDSSs) have become vital tools for organizations seeking to improve care delivery. CDSS tools have the ability to analyze large volumes of data and suggest the next steps for treatment, flagging potential problems and enhancing care team efficiency [3].

The information explosion via technology adoption and corresponding data collection on patients has re-oriented our approach to managing population health through the social determinants of health framework (a macro approach) on one hand and how to treat individual patients through precision medicine on the other (a micro approach). Indeed, artificial intelligence (AI) technology is necessary to achieve the goal of “precision medicine”, where medical decisions and treatments are tailored toward patient specifics. Precision medicine presupposes the availability of massive computing power and algorithms that can learn by themselves (machine learning) at an unprecedented rate [4]. An additional driver of AI technology is the sheer volume of healthcare data due to healthcare experiencing an information boom. The rapid expansion of scientific knowledge and the pace of technological development have resulted in an overwhelming sea of data that is difficult to decipher and apply [5].

Although precision medicine and social determinants of health (SDOH) address the same target (i.e., improving population health), each has been viewed as operating independently from the other, making the national goal of improving population health less effective and costlier. The purpose of this paper is to present an exploratory road map on the role of artificial intelligence in healthcare management. In this paper, AI is considered as an integrator/communicator between these two constructs: social determinants of health and precision medicine in population health.

This paper is organized as follows. A brief literature survey is presented in Section 2, where we discuss clinical–community linkages using AI as a communicator/integrator of the two health improvement constructs. Research methodology is discussed in Section 3. Based on the literature review, an exploratory roadmap in integrating AI into two health constructs (precision medicine and social determinants of health) is presented in Section 4. Discussion and limitations of this study of using AI in population health management are presented in Section 5. Conclusions are outlined in Section 6.

## 2. Literature Review

Widespread adoption of electronic health records (EHRs) through AI has resulted in the collection of massive amounts of clinical data. The growth of wearable medical devices and the predictive power of diagnostic information with multiple patient characteristics has made AI techniques a promising way to analyze these multimodal data sets [6]. However, extracting and analyzing the wealth of information in an accurate, timely, and reliable manner has been a continual challenge for healthcare providers. Accordingly, digital transformation in healthcare is a natural response to this challenge. Reddy [6] lists several reasons for the digital transformation in the healthcare arena: the rise of on-demand healthcare, the importance of big data in healthcare, treating patients by virtual reality, the growth of wearable medical devices, predictive healthcare management, the wonders of artificial intelligence, blockchain protection on personal and confidential patient information, and the promise of better electronic health records (EHRs). Though some electronic health record systems now allow clinicians to document certain social determinants of health in structured fields, these data are often missing or may be recorded in the format of free, unstructured text [7].

For digital health initiatives to more fully impact upon patient care, medical informatics specialists are investing resources in AI and machine-learning-based algorithms [8]. In some cases, machine-learning-based algorithms are allowing investigators to conduct a more in-depth subgroup analysis of previous published studies, the results of which call into question the practical application of the original investigation [8]. The digital transformation in the healthcare arena also opens a new frontier of on-demand healthcare. One in three American adults have gone online to assess a medical condition, 72% of internet users say they had looked online for health information within the past year, 47% of internet users search for information about doctors or other health professionals, and 38% of internet users search for information about hospitals and other medical facilities [9]. The most commonly searched topics are specific diseases or conditions, treatments or procedures, and doctors or other health professionals. Telemedicine, artificial intelligence (AI)-enabled medical devices, and blockchain electronic health records are just a few concrete examples of the digital transformation in healthcare which are completely reshaping how we interact with health professionals, how our data is shared among providers, and how decisions are made about our treatment plans and health outcomes.

On the other hand, the World Health Organization’s Report on Digital Health Strategy 2020–2025 [10] addresses population health using the digital transformation of health. WHO’s position appears to be much broader in scope in targeting population health and not just in precision medicine. The literature on AI in healthcare in medical intervention has been primarily limited to precision medicine [11,12,13]. Reasons might be that improving precision medicine may achieve population health. However, such a view addresses only a part of population health. Population health is considered more than the sum of health improvements made by precision medicine.

Population health requires investments in health-related areas in order to improve overall community health. Social determinants of health (SDOH) have been used as the main strategic tool to address community health. WHO defines SDOH as the conditions in which people are born, grow, live, work, and age. These circumstances are shaped by the distribution of power, money, and resources on global, national, and local levels [14]. The nation’s health, according to SDOH theory, depends on the conditions more than medical intervention (precision medicine). In fact, research shows that in order to improve population health, factors that impact the nation’s health should be addressed in addition to precision medical care. Research seems to indicate that the social determinants of health have a greater impact on a nation’s health than medical intervention. For example, the healthcare delivery system is responsible for only a fraction (about one fifth) of what keeps people healthy. Compared to medical services, SDOH have two times the responsibility for health outcomes [15]. Indeed, abundant research shows that the population health measured by infant mortality and life expectancy depends more on social determinants such as unemployment rates, alcohol, gun-related homicides, and insurance coverage than medical intervention [16]. However, Kim and Kwon’s study [16] did not address the role of AI in population health.

The Accountable Health Communities Model by the Centers for Medicare and Medicaid Services [17] lists a few of the largest drivers of health care costs that fall outside the clinical care environment (precision medicine). According to a CMS report, social and economic determinants, health behaviors, and the physical environment significantly drive utilization and costs. The evidence seems to be clear that addressing health-related social needs through clinical–community linkages can improve health outcomes and delivery costs.

Broad categories of social determinants of health include social factors, health services, individual behavior, and biology and genetics. It is the interrelationship among these factors that determines individual and population health. However, SDOH data is a messy proposition from the very first step of defining the social determinants of health. People define SDOH differently and inconsistently. Patients do not have “social determinants” per se; they have specific health-related social needs that impact their health and day-to-day lives. SDOH data is a complex collection of disparate data points, which causes difficulties when trying to make inferences [18]. Furthermore, social need screening requires information in the most private and potentially stigmatized areas of patients’ lives, including poverty, racism, intimate partner violence, etc. [19]. Only 16% of physicians and 24% of hospitals currently screen key social risk domains [20]. A lack of relevant and pertinent data/information on a patient’s SDOH could be one of the reasons for such a poor use of SDOH information during the assessment of a patient’s health status. AI could play clinical–community linkage roles, connecting with and supplementing information in precision medicine to SDOH information, making the overall health outcomes better and less costly.

However, it is unclear which social determinants of health best and most strongly relate to clinical and community health interventions, as the list of factors in SDOH is huge and in many instances uncollectible. To target specific communities for social intervention, some researchers developed a Social Deprivation Index (SDI) that specifically quantifies levels of disadvantage across small areas, evaluates their associations with health outcomes, and addresses health inequities. This measure of social deprivation, in combination with other indicators, has potential application in identifying areas that need additional health care resources [21]. Maroko et al. [22] saw deprivation as a state of observable and demonstrable disadvantage relative to the local community where an individual belongs. This disadvantage may occur at various levels as, for example, with regard to food, clothing, housing, education, or work. In fact, a person is considered deprived to the extent that they fall below the level attained by much of the population or below what is considered socially acceptable in the community.

The SDI is a complex index. Nevertheless, it will provide information at a local level where precision medicine can be effectively administrated. Maroko et al. [22] lists a working framework where SDI can be operationalized as shown in Table 1.

We postulate that medical intervention through SDOH produces better health outcomes (i.e., in a person’s health) with less cost if the consumers (patients) live in a community where social determinants are favorable for fostering community health. Yet, the literature appears to indicate that precision medicine and social determinants of health have developed independently from each other, although these two constructs are related and targeting the same objective: improving population health. A unifying communicator across the healthcare spectrum is needed to integrate these two health improvement tools to achieve the desired population health. We consider that AI should play a clinical–community linkage.

## 3. Research Methodology 

This paper discusses a literature review as a methodology for conducting this study. Knowledge production within the field of AI and precision medicine is accelerating at a tremendous speed, while at the same time remaining fragmented. Although AI and machine learning have been used extensively in industrial and supply chain fields, it is relatively new in the healthcare area to deploy AI and machine learning as a decision-making tool. It is only after the COVID-19 pandemic that people in the healthcare field started to pay attention to this decision-making tool. This makes it harder to keep up with the state-of-the-art and to be at the forefront of research, as well as to assess the collective evidence in a particular area of business research [23]. This is the one of reasons why the literature review as a research method is more relevant than before.

## 4. Developing an Exploratory Model

Based on a previous SDOH model by Kim and Kwon [16] and a literature review on clinical–community linkage, the role of digital application to health can be mapped as in Figure 1.

A nation’s health depends on the level of investment in three tracks: information infrastructure (hardware as well as workforce development), healthy communities (SDOH), and medical assets (precision medicine). AI is considered as an integrator to make information on patients and community health characteristics available to healthcare providers. We believe that in order to achieve a maximally healthy community, healthcare providers should and must have access to patient information not only for their personal medical history but also for the community characteristics where they live and interact. Imbalanced investment on any one of these tracks would not achieve population health at its full potential.

### 4.1. Investing in Health Technology Infrastructure

The literature appears to suggest that the investment of technologies in healthcare yields dividends in both cost savings and better health outcomes [24]. However, a rapid deployment of digital communication devices creates a “digital divide” between those who have access to technologies and digital literacy, and those who do not have access to such technologies. In healthcare, the digital divide can lead to disparities in patient portal adoption, telehealth care access, or the ability to utilize patient-facing management software such as online appointment schedulers. Nearly 75% of households either lack or are unaware of telehealth options, or both [25]. Black patients were four times more likely than white patients to visit the emergency department, not telehealth, during the pandemic’s initial surge [26]. Access to digital technologies assumes that every citizen can participate in the healthcare improvement process. Investment in digital infrastructure is a necessary first step toward achieving the goal of minimizing or eliminating the digital divide. One study shows that investment in key clinical health AI applications could create $150 million in annual savings for the U.S. healthcare economy by 2026 [24].

### 4.2. Investing in Social Determinants of Health

Data collection in social determinants of health (personal and household income and expenses by items; neighborhood characteristics; education; food security; and community characteristics including safety, social interaction on a personal basis, healthcare accessibility, etc.) should be easier with digital networks. However, this is a challenging task, as it involves and pertains to sensitive personal information. Since the Social Deprivation Index (SDI) requires information on an individual basis (not households or blocks) in very small communities of residence (blocks or streets in neighborhoods), people are reluctant to reveal sensitive information. For instance, information such as criminal record, alcohol addiction, numbers of marriages/divorces, child/spouse abuse, places they spend their income, welfare checks, etc. are personal, private, and pervasive. As a result, the information collected is messy, insufficient, or not reflective of the community characteristics. Nevertheless, the information listed above is essential in creating the Social Deprivation Index. There are a few studies that have successfully collected such information in creating the SDI [22].

### 4.3. Investing in Precision Medicine

The United States has invested an enormous amount of resources in medicine, drugs, and personnel workforce in this area, more than any other advanced country. The 2019 statistics show that this country spent $3649.4 billion on healthcare. Expenditures were, in part, $1191.8 billion on hospitals (32.7%), $564.4 billion on physicians (15.5%), $335 billion on drugs (9.2%), and $174.4 billion on investment (4.8%) [27]. It is interesting to note that the United States spent less than 5% of its healthcare spending on investment, while almost 60% of its spending was on hospitals, physicians, and drugs. It appears that there is room to increase the size of investment, including in information infrastructure and training the workforce in IT areas to make healthcare operations leaner, more efficient, and more effective without increasing healthcare spending.

### 4.4. Roles of AI as a Clinical–Community Integrator

The current healthcare practice in this country is that the two important health-enhancing constructs (SDOH and precision medicine) play separately and independently from each other [16]. We speculate that the improvement of population health requires that these two constructs work together to produce maximum health benefits. AI could and should play a role as an integrator/communicator between these two constructs. Since important healthcare information is now in the hands of healthcare providers, the social determinants of a patient should be merged with the patient’s health-specific information from their health records. AI by machine learning can generate and map the treatment protocol, and the sum of individual health would then be closer to the nation’s overall population health.

## 5. Discussion and Limitations

Although digital transformation in healthcare provides numerous benefits, there are issues and challenges in adopting AI and other technologies. For example, according to the 2018 Price Waterhouse Cooper survey, only 38% of chief executive officers of U.S. healthcare systems reported a digital component in their overall strategic plan [28]. Over 90% of respondents in the same survey pointed to data-protection and privacy regulations such as the 1996 Health Insurance Portability and Accountability Act, and the expansion of rules and penalties under the Health Information Technology for Economic and Clinical Health Act as factors limiting digital strategies at a micro level. Addressing the same issues of limiting the use of digital devices, Mehta et al. [1] cite three reasons: (1) reliability and availability of data, (2) matching AI tools to the right provider, and (3) an insufficient focus on ensuring that the workforce is motivated to use these tools effectively.

Crabb [29] broadens the limitations and challenges in using AI in healthcare industries. Many healthcare providers are still operating with technologies and data infrastructures that are incompatible with new, high-speed technologies. In addition, there is a conflict in reimbursement models (fee-for-service vs. value-based care). Despite some studies showing a positive return on investment, the evidence is limited. From a personnel standpoint, there is resistance to change with healthcare systems, especially from physician groups. Finally, existing studies have not addressed the impact of digital health on the overall healthcare ecosystem. The speed of technological innovation has outpaced the industry’s willingness and/or capability to adopt new technologies, leaving healthcare leaders further behind the roadmap to achieve their strategic goals. Investment in information infrastructure, including in workforce training, is the best way to achieve improved population health. Finally, this study has not addressed privacy issues in the application of AI to social determinants of health. One of the major issues that has hampered the collection of personal and private information is the fear that such information may invade a participant’s personal life. This issue must be addressed in a public forum, and some types of legislation may be needed to protect personal information before AI is fully utilized in healthcare fields.

## 6. Conclusions

Despite the many challenges, AI in healthcare industries will continuously play a significant role as we face four important new waves in the healthcare market: consumerism (on-demand health services), healthcare reform (health as a public good), efficiency (cost and process), and the patient–clinician–caregiver relationship (personal interaction with caregivers). We argue that investment in three areas (IT infrastructure, SDOH, and precision medicine) would be the best way to achieve better population health.

## Figures and Tables

**Figure 1 ijerph-18-05018-f001:**
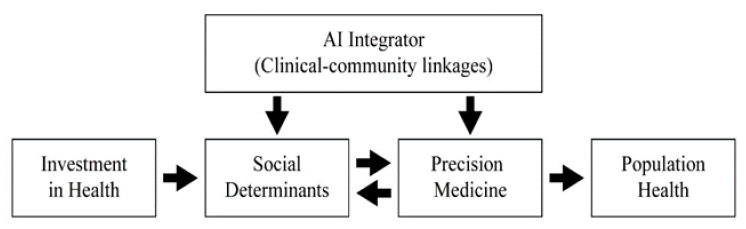
Roadmap to population health.

**Table 1 ijerph-18-05018-t001:** Factors in the Social Deprivation Index.

Domain	Variables
Education	% Population aged 25 years or older with less than 9 years of education
	% Population aged 25 years or older with at least a high school diploma
	% Employed population aged 16 years or older in white-collar occupations
Income/employment	Median family income in US dollars
	Income disparity
	% Families below federal poverty level
	% Population below 150% of federal poverty level
	% Civilian labor force population aged 16 years and older who are unemployed
Housing	Median home value in US dollars
	Median gross rent in US dollars
	Median monthly mortgage in US dollars
	% Owner-occupied housing units
	% Occupied housing units without complete plumbing
Household characteristics	% Single-parent households with children younger than 18
	% Households without a motor vehicle
	% Households without a telephone
	% Households with more than 1 person per room

Reproduced with permission from Maroko et al., 2016.

## Data Availability

Not applicable.

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
