# Peer review of "Integrating Social Determinants of Health to Precision Medicine through Digital Transformation: An Exploratory Roadmap"

_ijerph, 2021, doi:10.3390/ijerph18095018_

Round 1
Reviewer 1 Report
This paper aims to present an exploratory road map on the role of Artificial Intelligence (AI) in digital health through the lens of two major constructs: Social Determinants of Health (SDOH) and precision medicine. For this study, the authors have presented a brief literature review to depict a big picture of the digital transformation in e-health, including the current challenges addressing clinical-community linkage, precision medicine, and SDOH. In doing so, the authors also discussed the proposition of an exploratory roadmap composed of five major components: investment in health, social determinants, precision medicine, AI integrator, and population health. Lastly, they overview some challenges using AI in healthcare industries.
Overall, the article is well-written, and the significance of the content is unquestionable. Also, bringing SDOH to precision medicine in the light of AI approaches appears to be very promising and in line with recent research [1, 2, 3]. Indeed, this issue reveals to be even more important given the critical scenario imposed by the COVID-19 pandemic. However, despite the relevance of the paper's subject, we may notice that its scope and desired contribution need further theoretical and practical advances (especially in the context of a possible journal publication).
We may assume that this paper's main contribution relies on the exploratory road map bringing AI as a clinical-community integrator. Unfortunately, only one paragraph was dedicated to this issue. In line 235, the authors mention, “AI could and should play a role as an integrator/communicator between these two constructors” . This issue is precisely what a reader would expect to see in more detail, given the paper's aim. For example, how should AI play a clinical-community linkage in practice? This lack of thoroughness is also noticed in the remaining components of the model. Since a literature review supports the paper, a proper theoretical articulation is necessary. This superficiality is also seen in the challenges section, which evidences the lack of references (the authors mentioned prior studies' existence but did not cite and explain them).
In methodological terms, the authors did not provide any information about the literature review process, which poses a significant concern due to the scientific rigor guarantee. Moreover, the lack of a Related Work section makes it difficult to understand the present study's position regarding state-of-the-art, including the addressed research gap. One last insight that I would like to suggest concerns the role of GDPR in the proposed model, given the relevance of this matter to society and its pertinence for data-driven solutions.
I hope these comments are helpful for the authors, and I wish them all the best.
Minor issues:
- Integrating Social Determinants of Health to Precision Medicine Though Digital Transformation: An Exploratory Roadmap -> through;
- This title seems to be very generic considering the scope of the paper focused on AI
- Between 84 and 85 lines was a broken line;
- "Millions" instead of billions in the statistics (lines 223 and 224).
References:
[1] Seligman, B., Tuljapurkar, S., & Rehkopf, D. (2018). Machine learning approaches to the social determinants of health in the health and retirement study. SSM-population health, 4, 95-99.
[2] Galea, S., Abdalla, S. M., & Sturchio, J. L. (2020). Social determinants of health, data science, and decision-making: Forging a transdisciplinary synthesis. PLoS Medicine, 17(6), e1003174.
[3] Bompelli, A., Wang, Y., Wan, R., Singh, E., Zhou, Y., Xu, L., ... & Zhang, R. (2021). Social determinants of health in the era of artificial intelligence with electronic health records: A systematic review. arXiv preprint arXiv:2102.04216.
Author Response
Please see 3 attachments.
- Revised manuscript with red as suggested.
- Our reponse to Reviwer No. 1
- Our response to Reviewer No. 2

Reviewer 2 Report
Dear authors, receive my greetings.
Your manuscript is interesting, however, you must improve and clarify some relevant points:
- You must clarify the research problem in the literature
- You must introduce a methodology chapter to explain how you develop your research
- You must have a conclusion chapter and a limitation and future lines of research as well
- You are missing very relevant topics in your research. For instance:
- Ethics in heathcare practices, in particular when you use data. Here you have an excellent reference for use in your research: https://dx.doi.org/10.1002/mcda.1729
- Dynamic Capabilities, when you are moving you the way you delivery your service, purpose or mission, you must prepare your resources for that. Here you have an excellent reference for use in your research: https://dx.doi.org/10.1109/ICE.2019.8792629
All the best.
Good luck!
additional comments:
The manuscript is interesting but it miss some important components, as methodology, conclusion, future lines of research and "hot topics" that nowadays are very relevant in the research.
Round 2
Reviewer 1 Report
I appreciate that the authors have demonstrated attention in partially answering some of the suggestions raised. However, IMHO, adding one paragraph is not sufficient to cope with the lack of thoroughness noticed in the paper (even with the opportunity to improve the discussion after our revision).
The work aims to present an exploratory road map on the role of AI in healthcare management, but the practical and theoretical discussion is insufficient in terms of contributions.
I’ll respectfully remain with my position of rejecting this paper, given the reasons raised in my previous review.
Best regards.
Author Response
We answered all questions that Reviewer asked.
Please see the attachment with red color.

Reviewer 2 Report
Dear authors, thank you for the improvements, but I really considerer that the research cannot be finished without these very relevant topics considerer inside the scope of your research.
Good work.
Author Response
1.Identify research question: Our response. we added research issue in Line 58 to 61.
2. Research methodology: our response. We added additional section (Section 3) on this topic (Line178)
3. Conclusion: We added a section on Conclusion (Line 285)
4-1. Ethics in healthcare practice. We addressed this issue in Section 5, Discussion and Limitations.
4-2. Dynamic capabilities. We decided to leave it to other research related to this topic.
